# Total Lipids and Fatty Acids in Major New Zealand Grape Varieties during Ripening, Prolonged Pomace Contacts and Ethanolic Extractions Mimicking Fermentation

Emma Sherman [1], Muriel Yvon [2], Franzi Grab [2], Erica Zarate [3], Saras Green [3], Kyung Whan Bang [3] and Farhana R. Pinu [1],*

[1] Biological Chemistry and Bioactives Group, The New Zealand Institute for Plant and Food Research Limited, Auckland 1025, New Zealand; emma.sherman@plantandfood.co.nz
[2] Viticulture and Oenology Group, The New Zealand Institute for Plant and Food Research Limited, Blenheim 7201, New Zealand; muriel.yvon@plantandfood.co.nz (M.Y.); franziska.grab@plantandfood.co.nz (F.G.)
[3] School of Biological Sciences, University of Auckland, Auckland 1010, New Zealand; s.green@auckland.ac.nz (S.G.); kban028@live.com (K.W.B.)
* Correspondence: farhana.pinu@plantandfood.co.nz

**Abstract:** Despite the important roles of lipids in winemaking, changes in lipids during grape ripening are largely unknown for New Zealand (NZ) varieties. Therefore, we aimed to determine the fatty acid profiles and total lipid content in two of NZ's major grape varieties. Using gas chromatography–mass spectrometry, absolute quantification of 45 fatty acids was determined in Sauvignon blanc (SB) and Pinot noir (PN) grapes harvested at two different stages of ripeness. Lipid concentrations were as high as 0.4 g/g in seeds of both varieties, while pulp contained the least amount. Many unsaturated fatty acids were present, particularly in grape seeds, while skin contained relatively higher amounts of saturated fatty acids that increased throughout ripening. For both varieties, a significant increase in lipid concentration was observed in grapes harvested at the later stage of ripeness, indicating an association between lipids and grape maturity, and providing a novel insight about the use of total lipids as another parameter of grape ripeness. A variety-specific trend in the development and extraction of grape lipids was found from the analysis of the must and ethanolic extracts. Lipid extraction increased linearly with the ethanol concentration and with the extended pomace contact time. More lipids were extracted from the SB pomace to the must than PN within 144 h, suggesting a must matrix effect on lipid extraction. The knowledge generated here is relevant to both industry and academia and can be used to develop lipid diversification strategies to produce different wine styles.

**Keywords:** GC-MS; wine; ripening indices; lipid extraction; varietal differences; juice matrix



## 1. Introduction

Lipids are an important group of molecules that directly contribute to wine aroma development, and recent work has suggested they may also play a role in wine mouthfeel [1,2]. Lipids, including fatty acids, phospholipids, sterols and others have important biological functions in all cell types such as energy storage, cellular communication, biological process regulation and maintenance of cell membrane structure [3,4]. This group of metabolites is distributed within different grape tissues, particularly concentrating in the skin and seeds. A study of six different grape varieties revealed that lipid concentration ranged from 0.15% to 0.24% of fresh berry weight [5]. Grape seeds usually contain the highest proportion of lipids and are rich predominantly in mono- and polyunsaturated fatty acids. Grape skins contain a range of lipids that act as the main protective barrier to prevent evaporation of water and cellular contents [6].

Very little research has been carried out thus far to characterize lipids in different grape varieties compared with other primary metabolites (e.g., sugars, organic and amino

acids). As the grape juice matrix mostly consists of water, there was a misconception among scientists that lipids are not extracted from the pomace to the juice, especially in the production of white wines as no, or very little, skin contact is used in this process. However, a pilot study on New Zealand Sauvignon blanc (SB) juices showed that the total lipid concentration of grape juices can be as high as 2.8 g/L, with <15% available as free fatty acids [7]. Moreover, the SB juices sampled during commercial processing were shown to contain a diverse range of lipid species, including odd-numbered and hydroxy fatty acids, as well as other common saturated and unsaturated free and bound fatty acids. Quantitative data from a recently published study also showed that a variety of free fatty acids ranging from C6 to C24 are present in New Zealand SB juices [8]. Palmitic, stearic, linoleic, and γ-linolenic acids were the four most abundant free fatty acids detected in the 380 juice samples analyzed [7,8]. In another study, Arita, et al. [9] carried out comprehensive lipidome analyses of Pinot noir (PN) and Koshu grape berries and found clear differences in fatty acids and other lipid components. For example, at least 36 of 49 lipid components were significantly higher in PN skins than Koshu skins. PN skins also contained more lipids that had alkyl chains with >18 carbons, and a loss of C18:3 fatty acids during the ripening of Koshu grapes was observed, which may have been converted into (Z)-hex-3-enal, the precursor of C6-aroma compounds.

As grape pulp contains comparatively less lipid and fatty acids than grape skins and seeds, most of these compounds are extracted by the grape juice from the pressing of grapes and through prolonged skin/pomace contact. Studies have shown that extended pomace or skin contact and different pressing conditions alter the fatty acid composition of the grape juices either by increasing specific fatty acids (C18:2 and C18:3 fatty acids) or by reducing the amount of C6 compounds [10]. Comparison among different white and red varieties of grapes indicated a variety and grape tissue–specific differences in lipid composition [4]. For example, grape seeds usually contain ~60% unsaturated fatty acids, with linoleic acid being the predominant, and a high concentration of glycerophospholipids. Moreover, lignoceric acid was one of the main free saturated fatty acids in grape skin along with palmitic and stearic acids [4,11]. These results clearly highlight the diversity of grape lipids across grape varieties and their localization in different grape tissues.

Lipids and fatty acids play a significant role in wine yeast metabolism, particularly under anaerobic winemaking conditions. *Saccharomyces cerevisiae* and many other wine yeast strains are not able to produce fatty acids while growing in the absence of oxygen [12]. Under winemaking conditions, yeast cells are subjected to various stresses arising from osmotic pressure, ethanol toxicity and anaerobiosis. Lipids, specifically fatty acids, present in fermentation media then become an important source of nutrients that ensure optimum growth and fermentation performance of wine yeasts [13]. The literature evidence shows that lipids and fatty acids can modulate wine yeast metabolism, thus influencing the production of wine aroma compounds [14–18]. Pre-fermentative supplementation of common saturated and unsaturated fatty acids usually presents in grape juices significantly affected the growth and metabolism of wine yeasts, albeit in different ways [15,18]. Therefore, the availability of different fatty acids, even during the pre-inoculum preparation, causes significant variations in the yeast cells, thus modifying their metabolism and overall aroma production during winemaking.

In addition to modifying their metabolic activity, availability of fatty acids and other lipids changes the membrane composition of *S. cerevisiae* cells, thereby providing better protection against different stresses [19,20]. Supplemented fatty acids (e.g., palmitic and palmitoleic acids or mixture of fatty acids) are promptly consumed by wine yeasts and incorporated into the cell, allowing the yeast cells to be more viable in different fermentation conditions, which resulted in better survival and fermentation performance [21,22]. In addition to providing protection at low temperature, different lipids and fatty acids present in the yeast cell membrane proved to provide protection against ethanol toxicity during fermentation [23]. Modification and rearrangement of yeast cell lipid composition by changing the availability of lipid molecules in the exogenous media could be used to

produce different styles of wines from grape juices with modulated lipid and fatty acid contents. Additionally, recent research by Phan and Tomasino [24] found that the PN lipidome could be used to predict wine origin, showing that lipids persist in wines after fermentation, albeit at very low concentrations (<0.1%). Investigations into potential sensory impacts of lipids in finished wines found that phospholipids could induce a detectable increase in perceived viscosity of model wine [1]. However, attempts to increase the concentrations of lipids in real wines by adding yeast product were unsuccessful, and therefore the implications for mouthfeel perception unclear [25].

Most of the studies carried out on New Zealand wines investigated the influence of fatty acids on wine yeast metabolism and aroma production. The only study completed on the comprehensive lipidome of grape juices in New Zealand was on SB juices [7]. Therefore, there is a lack of knowledge about lipid composition in different grape varieties and how these molecules contribute to the wine quality and sensory properties. This project aimed to fill the gap in knowledge, mainly by determining total lipid content and fatty acid composition of two main New Zealand grape varieties. Here, we investigated the differences in lipid and fatty acid composition at different stages of ripening of SB and PN grapes while determining the effect of skin (pomace) contact time on lipid extraction into the must. We also explored the influence of alcohol concentration (mimicking wine fermentation) on lipid extraction from grape pomace.

## 2. Materials and Methods

### 2.1. Experimental Design-Collection of Grapes and Sampling Protocols

Approximately 30 kg of grapes was harvested from the same blocks at two different ripening stages (harvest 1 = unripe grapes and harvest 2 = ripe grapes). SB was sourced from the Marlborough Research Centre's Rowley Vineyard (harvest 1 on 23 March 2019 and harvest 2 on 30 March 2019) while PN grapes were harvested from Omaka Settlement Vineyard of Dog Point situated in Marlborough wine region, New Zealand (harvest 1 on 15 March 2019 and harvest 2 on 22 March 2019). These vineyards follow the standard seasonal canopy management used in New Zealand.

After harvest, a representative sample of 100 grapes was collected from approximately 25 different bunches from each harvest to separate and collect the different grape tissue types (skin, seeds, and pulp). Then, 25 kg of grapes was weighed, crushed, and destemmed using dry ice and 100 ppm potassium metabisulfite (PMS) was added. Using a hydro-bladder press (Fratelli Marchisio & C.S.p.A., Pieve di Teco, Italy), crushed grapes were immediately pressed following protocols for SB developed in our research winery and the juice collected under $CO_2$ cover. Resulting juices and pomaces were weighed separately and the juice-to-pomace ratio was determined for the reconstitution of the musts and pomaces. A further 60 ppm PMS was added to each treatment to minimize oxidation. Figure 1 shows a schematic diagram of the sample preparation protocol (total $n$ = 122 including grape skin, seeds, and pulp). Approximately 100 mL fresh juice after each press was collected in triplicate for various analyses. Juices and pomaces were reconstituted with the same juice and pomace ratio determined after pressing (final volume ~2 L) in triplicate to simulate extended pomace contact under standard cold soak conditions at 6 °C. Pomace contact time varied depending on grape variety to make it relatable with commercial winery practices: SB juices were kept in contact with pomace for 12 and 24 h while PN juices were in contact with pomace for 72 and 144 h. Another set of samples was prepared by reconstituting pomace using aqueous ethanol solutions in triplicate (final volume ~2 L) using the same juice and pomace ratio after pressing at three different ethanol concentrations (0%, 9% and 13% ethanol). Moreover, 0% mimicked the beginning of fermentation, while 9% and 13% represented mid and late stages of fermentation. Pomace contact was maintained for 72 and 168 h; however, the temperature regime was different for the grape varieties. SB ethanolic extractions of pomaces were carried out at 15 °C resembling commercial SB winemaking in New Zealand and PN ethanolic extractions of pomaces were performed at 24 °C, similar to commercial PN production (Figure 1).

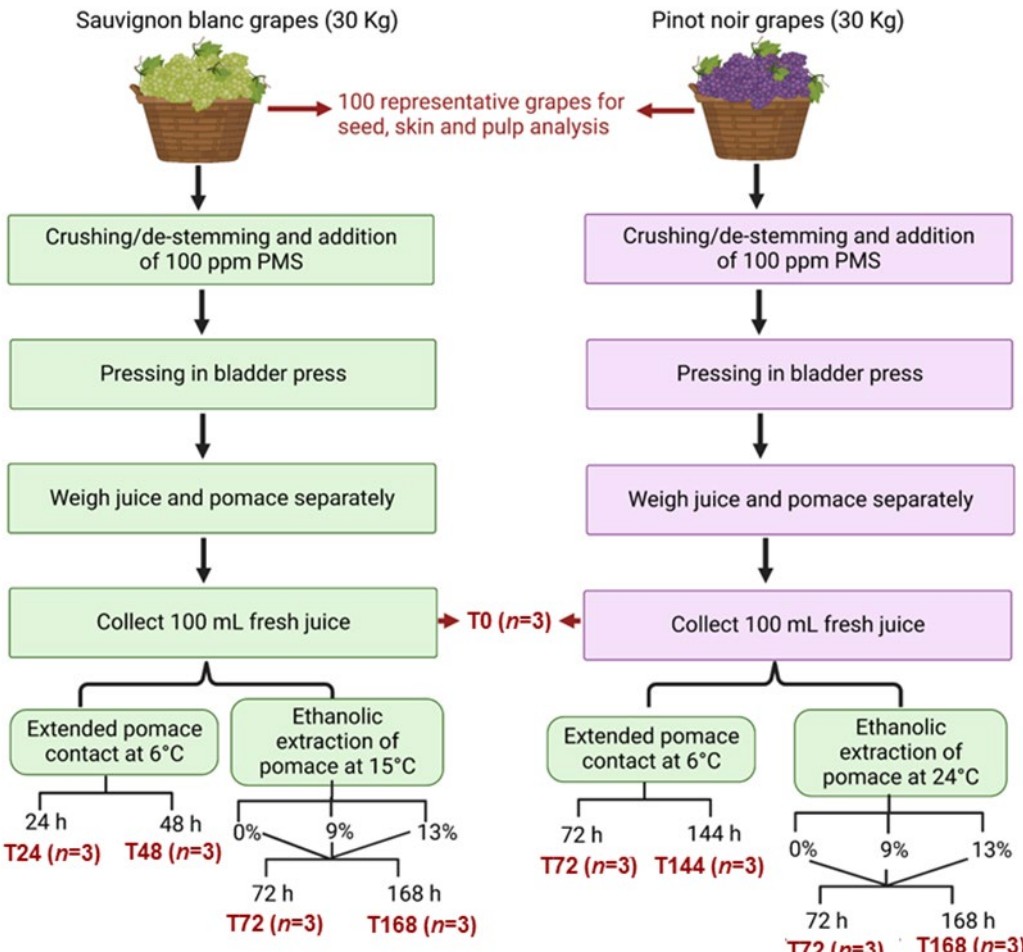

**Figure 1.** Sample preparation protocol for the project. Here, PMS = potassium metabisulphite, T0 = fresh juices without pomace contact. T denotes the time of pomace contact, number of samples from each stage is shown in red. Green represents Sauvignon blanc while purple shows Pinot noir sampling method. The total number of samples was 122.

*2.2. Determination of Major Oenological Parameters*

Total soluble solids (TSS) content of all samples was determined using a handheld digital refractometer (Atago, Tokyo, Japan). Titratable acidity and pH were determined using a Mettler Toledo (Columbus, OH, USA) T70 autotitrator with an end-point titration to pH 8.2 and calculated in tartaric acid equivalents (g/L) [26]. Aqueous sodium hydroxide (0.1 M) was used as the titrant. Wine samples were degassed prior to analysis.

Primary amino acid (PAA) concentrations were measured using the nitrogen by o-phthaldialdehyde (NOPA) assay adapted for small volumes [27]. The reaction was measured using a Molecular Devices (San Jose, CA, USA) Spectramax 384 Plus plate reader with a 1-cm pathlength cuvette reference correction. Sample PAA concentrations were quantified in duplicate against a five-point isoleucine standard curve ($R^2 > 0.98$). Ammonium concentrations were measured by an enzymatic assay monitoring the deprotonation of NADPH at 340 nm using the plate reader. Enzymes were purchased from Megazyme (Bray, Ireland); ketoglutaric acid was purchased from Sigma-Aldrich (St. Louis, MO, USA). Samples were appropriately diluted (usually two-fold) and quantified in duplicate against a five-point standard curve ($R^2 > 0.98$). Yeast available nitrogen (YAN) was calculated as the sum of PAA plus ammonium expressed in mg/L of nitrogen.

Glucose and fructose were quantified by enzymatic assay based on the reduction in NADP to NADPH, and the reaction was monitored at 340 nm using the plate reader.

Enzymes and cofactors were purchased from Megazyme (Bray, Ireland). Samples were appropriately diluted and quantified in duplicate against an eight-point standard curve ($R^2 > 0.98$). This method was adapted from the Compendium of International Methods of Analysis OIV-MA-AS311-02.

The optical density of the juice was determined in duplicate directly in a UV-transparent 96-well microplate at 280, 320, and 420 nm based on the method described in Martin, et al. [26]. Absorbance at 280 nm was used to quantify total phenolics against a five-point gallic acid standard curve (five-point, $R^2 > 0.98$).

### 2.3. Quantification of Major Organic and Amino Acids

A Shimadzu Prominence high performance liquid chromatography (HPLC) (Shimadzu Corporation, Kyoto, Japan) system with a diode array detector (DAD) equipped with an Allure Organic Acids Restek column (Bellefonte, PE, United States; 5 μm, 240 × 4.6 mm) was used to quantify organic acids (tartaric, malic, ascorbic, citric and succinic acids) in samples based on the method reported by Shi et al., 2011 [28] and validated in our laboratory. Briefly, samples were diluted five-fold and filtered through a 0.22 μm syringe filter before injection. A 25-min isocratic method using phosphate buffer (25 mM, pH 2.3) was run at 0.6 mL/min and 30 °C. Samples were run in duplicate and quantified on a six-point standard curve ($R^2 > 0.98$).

Quantification of amino acids in grape juices was performed on an Agilent 1200 series HPLC (Santa Clara, CA, USA) equipped with a Thermo Fisher Scientific Hypersil Gold C18 column (Waltham, MA, USA; 5 μm, 250 × 3.0 mm) using gradient elution with a phosphate/borate buffer (10 mM each, pH 8.2) containing 0.1 *v/v*% tetrahydrofuran (THF) and a combination of methanol, acetonitrile and water as the organic solvent (45:45:10), run at 1.5 mL/min and 40 °C. The method was adapted from an application note by Henderson and Brooks [29], validated in the laboratory and described in Martin et al. [26,27]. Briefly, online derivatization of primary amino acids was carried out with o-phthalaldehyde and 3-mercaptopropionic acid and detected by a DAD at 340 nm excitation and 450 nm emission. Samples were treated with iodoacetic acid to encourage the reduction in cysteine. Secondary amino acids were derivatized online with 9-fluorenylmethyl chlo-roformate and detected by fluorescence (260 nm excitation, 315 nm emission). A standard mix of 17 amino acids was purchased from Agilent (Santa Clara, CA, USA). Internal standards sarcosine (100 mg/L) and α-aminobutyric acid (100 mg/L) were added to all standards and samples to account for potential injection volume variability. Samples were analyzed both undiluted and diluted ten-fold in water for the quantification of low and high abundance amino acids respectively and filtered through a 0.45-μm syringe filter before injection. All samples were analyzed in duplicate and quantified on a six-point standard curve ($R^2 > 0.98$) [29].

### 2.4. Lipid Extraction and Transesterification

Lipid extraction of the samples was performed using a modified protocol published by Díaz de Vivar, et al. [30]. As different types of samples were generated during this study, optimization of the extraction method was carried out to determine the amount and volume of sample required for extracting lipids. For juice and alcoholic extracts, 2 mL of the sample provided optimum results while 10 mg of the sample was required for analyzing seeds, skin, and pulp. Grape seeds, skins and pulp were freeze-dried and ground into a powder, which was then used for the extraction of lipids and fatty acids. Lipids were extracted by adding 250 μL of distilled water to 2 mL of juice and alcoholic extracts and 10 mg powder of seeds, skins and pulp. After this, 125 μL of chloroform with internal standard (C23: 0.48 mg in 50 mL of chloroform) and 250 μL of methanol were added and mixed thoroughly. The samples were centrifuged at 2500 rpm for 5 min and transferred to new glass vials. Another 250 μL of distilled water and 250 μL of chloroform were added to the samples, mixed, and again centrifuged at 2500 rpm for 5 min. The top layer was discarded and the bottom layer containing the lipids in chloroform was transferred to a gas-chromatography

(GC) vial. Using a SpeedVac (Savant SP5121P, ThermoFisher, Waltham, MA, USA) the chloroform layer was concentrated prior to transesterification.

Transesterification was performed using a modified protocol published by Díaz de Vivar, et al. [30]. Lipid extract (1 mL) was transferred to screw-cap glass culture tubes and 2 mL of internal standard (C19:0) solution dissolved in methanol: toluene 4:1 (*v/v*) was added. For the seed samples, a 1:10 dilution was performed using toluene. A magnetic stir bar was placed inside the tube and 200 μL of acetyl chloride was added to the tube over a period of 1 min. The tube was tightly closed with the Teflon cap prior to determining the weight and it was then placed in a heating/stirring dry block at 100 °C for 1 h. After this, the tube was cooled in water, dried and the weight was determined again to check for any leakage. To stop the reaction and neutralize the mixture, 5 mL/sample of 6% $K_2CO_3$ solution was added very slowly and mixed by vortexing prior to centrifuging at 2500 rpm for 5 min. Approximately 200 μL of the upper toluene phase was transferred to a vial for analysis by gas chromatography–mass spectrometry (GC-MS).

*2.5. GC-MS Analysis*

The transesterified lipid extracts were injected on to a GC-MS (Agilent GC 7890 coupled to a MSD 5975, Agilent Technologies, Santa Clara, CA, USA) with a quadrupole mass selective detector (EI) operated at 70 eV using helium as the carrier gas. Instrument analytical parameters were based on those developed by Kramer et al. [31]. Column selection was based on the recommendations from the Official Methods for the determination of trans fat (American Oil Chemists Society). The column was a fused silica Rtx-2330 100 m long, 0.25 mm internal diameter, 0.2 μm highly polar stationary phase (90% biscyanopropyl 10% cyanopropylphenyl polysiloxane, Shimadzu, Kyoto, Japan). The carrier gas was instrument grade helium (99.99%, BOC). One microlitre of the sample was injected using a CTC PAL autosampler into a glass 4-mm ID straight inlet liner packed with deactivated glass wool (Restek Sky®, Bellefonte, PA, USA). The inlet temperature was 250 °C, in splitless mode, and the column flow was set at 1 mL/min, with a column head pressure of 9 psi, giving an average linear velocity of 19 cm/s. Purge flow was set to 50 mL/min 1 min after injection. After injection at 60 °C, the oven temperature was raised to 150 °C at a rate of 40 °C min$^{-1}$, and then to 230 °C at 3 °C min$^{-1}$ and finally held constant for 30 min. The GC oven temperature programming started isothermally at 45 °C for 2 min, increased 10 °C/min to 215 °C, held 35 min, increased 40 °C/min to 250 °C and held 10 min. The transfer line to the mass spectrometric detector (MSD) was maintained at 250 °C, the MSD source at 230 °C and the MSD quadrupole at 150 °C. The detector was turned on 14.5 min into the run. The detector was run in positive-ion, electron-impact ionization mode, at 70 eV, with the electron multiplier set with no additional voltage relative to the autotune value. Data were acquired at 1463 amu/s in scan mode from 41 to 420 atomic mass units, with a detection threshold of 100 ion counts.

Fatty acid methyl ester (FAME) peaks were identified by comparing their retention times with those of 52 authentic FAME standards (Nu-Chek-Prep, Inc., Elysian, MN, USA-52 component FAME mix; GLC reference standard 674) and based on the in-house MS library. A five-point calibration curve was prepared in order to quantify the fatty acids positively identified in the samples.

*2.6. GC-MS Data Mining*

Data analysis was automated and performed with in-house R package developed at the University of Auckland metabolomics laboratory [7,8]. The raw data output from the GC-MS was converted to AIA format (.cdf) and analyzed using automated mass spectral deconvolution and identification software (AMDIS, http://www.amdis.net/, accessed on 4 April 2020) and an in-house MS library of 52 fatty acids derivatized compounds and one extra standard (C19:0). The reference ion used as a measure of abundance for each compound is usually the most abundant fragment and is not the molecular ion. As the output from AMDIS returns zero values that are not suitable for statistical analysis, an

in-house R-script MassOmics was used in conjunction with the AMDIS output to produce data that include trace levels of metabolites normally excluded by AMDIS. The values are generated from the maximum height of the reference ion for the compound peak. Unlike peak area, peak height is affected by chromatographic disturbances such as column contamination, and as a result early eluting peaks may sometimes be under-represented. Data were checked against negative controls and obvious contamination, or artifacts were highlighted in the uncorrected results and removed in the corrected results. Coeluting peaks were highlighted, checked and corrected. Where two identifications were equally likely for one peak, both identifications have been reported. The resulting dataset was then normalized by the internal standard "nonadecanoic acid" and average peak responses from experimental "blank" samples were deducted from experimental samples to account for baseline response. Quantification was performed using calibration curves obtained from the analysis of standard mix samples. Lastly, sample biomass/volume normalization and dilution correction (if required) were performed to obtain the final quantified data. Approximate total lipid content was calculated by summing the concentrations of all detected free fatty acids in the samples according to Van Wychen and Laurens [32].

*2.7. Statistical Analysis*

Data were log-transformed prior to performing any statistical analyses. Independent Student's *t*-tests were performed to compare the changes in the total lipids, fatty acids and oenological parameters (e.g., alcohol content, volatile thiol concentrations, amino and organic acids) of each treatment compared with their respective controls using an in-house R script. One-way analysis of variance (ANOVA) was also performed to compare different treatments. Microsoft Excel 2010 was used to determine mean and standard deviation of triplicate controls and treatments. A web-based platform Metaboanalyst 4.0 (http://www.metaboanalyst.ca, accessed on 4 April 2020) was used to perform different unsupervised and supervised statistical analyses including principal component analysis (PCA), partial least square-discriminant analysis (PLS-DA), variance in projection (VIP) and also to generate the heatmap (distance measure: Euclidean and clustering algorithm: Ward). A machine learning algorithm in Metaboanalyst 4.0 "pattern searching" was also used to determine the top 25 features correlated (using Pearson correlation method) with the total lipids and major fatty acids [33].

## 3. Results and Discussion

*3.1. Chemical Composition of the Sauvignon Blanc (SB) and Pinot Noir (PN) Juices and Alcoholic Extracts*

We determined the chemical composition of both the SB and PN juices just after pressing, juices with prolonged pomace contact and ethanolic extracts of the grape pomaces. Table 1 details the main oenological measurements and organic and amino acids present in the SB and PN juices. Although we intended to collect the SB grapes at 18 °Brix for harvest 1 and 21.5 °Brix for harvest 2, the TSS in our final samples was 19.9 °Brix and 21.8 °Brix, respectively. Similarly, the TSS in the PN grapes was 22.1 °Brix (instead of 20 °Brix) and 24.7 °Brix (instead of 23.5 °Brix) for harvest 1 and harvest 2, respectively. As the Marlborough region experienced a much warmer summer in 2019, the grape ripening was faster than previous vintages. However, there was at least a 2 °Brix difference in the TSS between the harvests, and titratable acidity and YAN data also reflected the different fruit maturities at each harvest (Table 1).

Most of the chemical parameters shown in Table 1 varied between harvests, depending on the grape variety. We observed no significant variation in pH and ammonium ($p > 0.05$) between harvest 1 and 2 in both the SB and PN juices. In the PN juices, both tartaric and malic acid concentrations were significantly lower and sugar concentrations were much higher in the harvest 2 juices. Tartaric acid and total reducing sugar concentrations changed little between harvests in the SB juices, but a significant reduction in malic acid concentration and titratable acidity was observed in the harvest 2 juices, suggesting a more

advanced ripeness stage compared to with harvest 1. SB and PN amino acids showed varietal differences, and most of their concentrations increased in the harvest 2 samples (Table 1). Additionally, total polyphenols increased 3.5-fold in the PN harvest 2 grape juices compared with harvest 1, suggesting that the harvest 2 grapes were indeed more mature.

**Table 1.** Chemical properties of harvest 1 and 2 Sauvignon blanc and Pinot noir juices prior to pomace contact.

| | Sauvignon Blanc | | | Pinot Noir | | |
|---|---|---|---|---|---|---|
| | **Harvest 1** | **Harvest 2** | *p*-Value | **Harvest 1** | **Harvest 2** | *p*-Value |
| Total soluble solids (°Brix) | 19.9 (0.1) | 21.8 (0.0) | <0.05 | 22.1 (0.1) | 24.7 (0.1) | <0.05 |
| Titratable acidity (g/L) | 11.7 (0.04) | 9.77 (0.02) | <0.05 | 9.09 (0.06) | 6.95 (0.05) | <0.05 |
| pH | 3.02 (0.01) | 3.05 (0.00) | >0.05 | 3.12 (0.01) | 3.22 (0.01) | >0.05 |
| Tartaric acid (g/L) | 7.66 (0.14) | 7.09 (0.02) | >0.05 | 6.30 (0.07) | 3.30 (0.08) | <0.01 |
| Malic acid (g/L) | 5.74 (0.11) | 4.70 (0.01) | <0.05 | 4.68 (0.05) | 3.62 (0.01) | <0.05 |
| YAN (mg N/L) | 155 (14) | 179 (4) | <0.05 | 191 (3) | 219 (0) | <0.05 |
| Ammonium (mg N/L) | 61 (14) | 67 (2) | >0.05 | 82 (0) | 81 (1) | >0.05 |
| PAA (mg N/L) | 94 (0) | 112 (2) | <0.05 | 109 (2) | 139 (1) | <0.03 |
| Total reducing sugar (g/L) | 221 (19) | 237 (4) | >0.05 | 231 (1) | 293 (11) | <0.05 |
| Glucose (g/L) | 122 (7) | 129 (3) | >0.05 | 121 (2) | 155 (7) | >0.05 |
| Fructose (g/L) | 99 (11) | 108 (7) | <0.05 | 110 (3) | 138 (4) | <0.01 |
| Total Phenolics (mg gallic acid/L) | * | 252 (2) | ND | 424 (5) | 1517 (69) | <0.01 |
| Aspartic acid (µmol/L) | 237 (7) | 397 (3) | <0.01 | 254 (3) | 161 (4) | <0.01 |
| Glutamic acid (µmol/L) | 827 (14) | 861 (14) | <0.05 | 751 (6) | 786 (15) | >0.05 |
| Serine (µmol/L) | 281 (5) | 357 (6) | >0.05 | 377 (5) | 504 (12) | <0.01 |
| Arginine (µmol/L) | 1243 (22) | 1476 (37) | <0.05 | 1473 (20) | 2143 (51) | <0.01 |
| Alanine (µmol/L) | 682 (11) | 806 (7) | <0.01 | 807 (8) | 1181 (21) | <0.01 |
| Histidine (µmol/L) | 810 (12) | 901 (17) | <0.05 | 767 (32) | 1102 (29) | <0.01 |
| Threonine (µmol/L) | 454 (6) | 446 (4) | >0.05 | 551 (9) | 682 (10) | <0.05 |
| Valine (µmol/L) | 143 (4) | 158 (6) | >0.05 | 153 (8) | 171 (4) | >0.05 |
| Proline (µmol/L) | 873 (107) | 1032 (93) | <0.05 | 681 (92) | 636 (54) | >0.05 |
| Methionine (µmol/L) | 77 (6) | 106 (2) | <0.05 | 142 (2) | 213 (4) | <0.01 |
| Isoleucine (µmol/L) | 48 (2) | 59 (1) | >0.05 | 89 (1) | 107 (2) | >0.05 |
| Leucine (µmol/L) | 51 (6) | 72 (1) | >0.05 | 124 (0) | 168 (3) | <0.05 |
| Phenylalanine (µmol/L) | 39 (0) | 64 (1) | <0.05 | 45 (0) | 59 (1) | >0.05 |

* indicates missing data; ND = not determined; PAA = primary amino acids; YAN = yeast available nitrogen. *p*-values were determined by comparing harvest 1 and harvest 2 values using independent *t*-test. Standard deviations are shown within brackets.

Chemical analyses of the musts from extended pomace contact (24 and 48 h contact for SB; 72 and 144 h contact for PN) revealed that the composition of the musts changed because of pomace contact, with the duration of contact an important factor in these variations. There was a significant reduction in total reducing sugars after 24 h of pomace contact in the harvest 1 SB must, which was not expected. This may be due to fermentation starting despite the higher $SO_2$ addition rates and cold temperature, or insufficient precision of analysis of high sugar concentration samples by the plate reader assay [34].

Concentrations of polyphenols, tannins and monomeric anthocyanins increased significantly ($p < 0.05$) in the PN musts because of extended pomace contact. As winemakers are aware, these secondary metabolites are extracted better with time during pomace contact, which is also widely reported in the literature, for example by [35].

### 3.2. Overview of Fatty Acids Present in Grapes and Their Extracts Harvested at Different Time Points

The main aim of our project was to determine the lipid content and composition in two main New Zealand grape varieties, and to explore how pomace contact and ethanolic extraction (mimicking winemaking conditions) influence lipid and fatty acid composition.

This is the first study of this kind in New Zealand, which has allowed us to generate unique datasets on SB and PN grape tissues, juices, and extracts. The method we used here provided an overview of fatty acids that are found in both bound and free form. Table 2 shows the complete list of fatty acids detected in all the samples. Using an in-house MS library, 45 fatty acids were positively identified and quantified in grape juices and extracts while 44 fatty acids were determined in grape seeds, skin, and pulp (Table 2). These fatty acids ranged from C8 to C24, the majority of which were unsaturated. This was in accordance with previously published data [4,7]. At least two medium-chain fatty acids, undecanoic and tridecanoic acids, were unique to grape skin and seeds, while four other long-chain fatty acids including oleic, trans- vaccenic, trans-elaidic and 11-cis-eicosenoic acids were only found in the grape juices and extracts. Therefore, either these long-chain fatty acids were too low in concentration to be detected in the grape tissues, or somehow, they were produced as a result of the extraction process. It is possible that some lipid and fatty acid oxidation occurred during the extraction process despite our measures to exclude oxygen (i.e., using $N_2$ gas prior to closing the vessels and adding 60 ppm PMS).

**Table 2.** List of saturated and unsaturated fatty acids detected and identified in grape juices, extracts and tissues.

| | Fatty Acids | Other Known Names | No of Carbons and Double Bonds | Type of Fatty Acid |
|---|---|---|---|---|
| 1 | Octanoic acid | Caprylic acid | C8:0 | Saturated |
| 2 | Decanoic acid | Capric acid | C10:0 | Saturated |
| 3 | Undecanoic acid * | Undecylic acid | C11:0 | Saturated |
| 4 | Dodecanoic acid | Lauric acid | C12:0 | Saturated |
| 5 | Tridecanoic acid * | Tridecylic acid | C13:0 | Saturated |
| 6 | Tetradecanoic acid | Myristic acid | C14:0 | Saturated |
| 7 | Pentadecanoic acid | | C15:0 | Saturated |
| 8 | Hexadecanoic acid | Palmitic acid | C16:0 | Saturated |
| 9 | Heptadecanoic acid | Margaric acid | C17:0 | Saturated |
| 10 | Octadecanoic acid | Stearic acid | C18:0 | Saturated |
| 11 | Eicosanoic acid | Arachidic acid | C20:0 | Saturated |
| 12 | Heneicosanoic acid | Heneicosylic acid | C21:0 | Saturated |
| 13 | Docosanoic acid | Behenic acid | C22:0 | Saturated |
| 14 | Tetracosanoic acid | Lignoceric acid | C24:0 | Saturated |
| 15 | 9-cis-Tetradecenoic acid | Myristoleic acid | C14:1 | Unsaturated |
| 16 | 9-trans-Tetradecenoic acid | Myristelaidic acid | C14:1 | Unsaturated |
| 17 | 10-cis-Pentadecenoic acid | | C15:1 | Unsaturated |
| 18 | 10-trans-Pentadecenoic acid | | C15:1 | Unsaturated |
| 19 | 9-cis-Hexadecenoic acid | Palmitoleic acid (cis) | C16:1 | Unsaturated |
| 20 | (E)-9-hexadecenoic acid | Palmitoleic acid (trans) | C16:1 | Unsaturated |
| 21 | 10-cis-Heptadecenoic acid | | C17:1 | Unsaturated |
| 22 | 10-trans-Heptadecenoic acid | | C17:1 | Unsaturated |
| 23 | 9-trans-Octadecenoic acid+ | Elaidic acid (trans) | C18:1 | Unsaturated |
| 24 | 9-cis-Octadecenoic acid+ | Oleic acid | C18:1 | Unsaturated |
| 25 | 11-trans-Octadecenoic acid+ | trans-Vaccenic acid | C18:1 | Unsaturated |
| 26 | 11-cis-Octadecenoic acid | cis-Vaccenic acid | C18:1 | Unsaturated |
| 27 | 9,12,15-cis-Octadecatrienoic acid | alpha-Linolenic acid | C18:3 | Unsaturated |
| 28 | 9,12-cis-Octadecadienoic acid | Linoleic acid | C18:2 | Unsaturated |
| 29 | 9,12-trans-Octadecadienoic acid | Linolelaidic acid | C18:2 | Unsaturated |
| 30 | cis-6,9,12-octadecatrienoic acid | gamma-Linolenic acid | C18:3 | Unsaturated |
| 31 | 10-trans-Nonadecenoic acid | Nonadecylic acid | C19:1 | Unsaturated |
| 32 | 7-trans-Nonadecenoic acid | | C19:1 | Unsaturated |
| 33 | 11,14,17-cis-Eicosatrienoic acid | Eicosatrienoic acid | C20:3 | Unsaturated |
| 34 | 11,14-cis-Eicosadienoic acid | Eicosadienoic acid | C20:2 | Unsaturated |
| 35 | 11-trans-Eicosenoic acid | Eicosenoic acid | C20:1 | Unsaturated |

**Table 2.** *Cont.*

|  | **Fatty Acids** | **Other Known Names** | **No of Carbons and Double Bonds** | **Type of Fatty Acid** |
|---|---|---|---|---|
| 36 | 11-cis-Eicosenoic acid+ |  | C20:1 | Unsaturated |
| 37 | 8,11,14-cis-Eicosatrienoic acid |  | C20:3 | Unsaturated |
| 38 | 5,8,11,14,17-cis-Eicosapentaenoic acid |  | C20:5 | Unsaturated |
| 39 | 5,8,11,14-cis-Eicosatetraenoic acid |  | C20:4 | Unsaturated |
| 40 | 13,16-cis-Docosadienoic acid |  | C22:2 | Unsaturated |
| 41 | 13-cis-Docosenoic acid |  | C22:1 | Unsaturated |
| 42 | 13-trans-Docosenoic acid |  | C22:1 | Unsaturated |
| 43 | 7,10,13,16,19-docosapentaenoaic acid |  | C22:5 | Unsaturated |
| 44 | 4,7,10,13,16,19-Docosahexaenoic acid |  | C22:6 | Unsaturated |
| 45 | 7,10,13,16-cis-Docosatetraenoic acid | Docosapentaenoic acid | C22:4 | Unsaturated |
| 46 | 4,7,10,13,16-docosapentaenoaic acid |  | C22:5 | Unsaturated |
| 47 | 15-cis-Tetracosenoic acid |  | C24:1 | Unsaturated |

\* indicates the fatty acids were only present in grape skin and seed; + indicates the fatty acids were only found in grape juices and extracts.

### 3.3. Total Lipids and Fatty Acids in Sauvignon Blanc and Pinot Noir Grape Tissues

Determination of the total lipids from transesterification of lipids to fatty acid methyl esters is a common and widely used analytical practice [36,37]. In grapes, different classes of lipids are distributed among the different tissue types. Therefore, we used this transesterification protocol to accurately quantify 45 fatty acids in the various grape tissues. The summation of all the fatty acids presents in the sample provided us an indication of the total lipid content of the samples [7]. As shown in Table 3, grape pulp contained the least amount of lipid as expected. Total lipid content was significantly higher in the PN pulp ($p < 0.05$) and skins ($p < 0.01$) than the SB from both harvests, while seeds from both varieties had similar amounts of lipids. However, grape lipids were mostly concentrated in seeds for both grape varieties, which was expected [4]. Lipids and fatty acids are usually extracted to the grape must during pressing and from subsequent skin/pomace contact [10].

**Table 3.** Saturated and unsaturated fatty acids detected and identified in Sauvignon blanc (SB) and Pinot noir (PN) juices, extracts and tissues. Standard deviations are shown within brackets.

|  | **Pulp (g/g)** | **Seeds (g/g)** | **Skin (g/g)** |
|---|---|---|---|
| SB harvest 1 | 0.009 (0.001) | 0.311 (0.075) | 0.013 (0.004) |
| SB harvest 2 | 0.016 (0.008) | 0.486 (0.089) | 0.023 (0.001) |
| Comparison (fold-change, SB harvest 1 vs. harvest 2) | 1.84 | 1.83 | 1.56 |
| PN harvest 1 | 0.016 (0.005) | 0.221 (0.025) | 0.025(0.007) |
| PN harvest 2 | 0.029 (0.008) | 0.438 (0.077) | 0.072 (0.009) |
| Comparison (fold-change, PN harvest 1 vs. harvest 2) | 1.78 | 1.99 | 2.92 |

Our data show an interesting trend of increasing lipids (1.5- to 3-fold) in the harvest 2 grapes for both varieties, indicating that lipid concentrations increase as grapes mature (Table 3). Many studies discuss the role of sugars and organic and amino acids in determining the ripeness of grapes [37–41], but the development of lipids and fatty acids during grape ripening is largely unknown. Our data provide novel information on how lipids can be strongly related to grape maturity and ripeness. It is noteworthy that traditionally used ripeness parameters including total soluble solids and titratable acidity are not necessarily linked with improving wine aroma and quality. However, lipids, specifically fatty acids, play a vital role during wine fermentation and particularly can contribute to the productions of various aroma compounds [15–18]. Therefore, if the total lipids or a specific lipid (fatty acid) marker are directly related with grape ripeness, this will provide researchers or even industries an option to make winemaking decisions, allowing them to diversify wine styles.

As expected, seeds from both varieties contained much higher amounts of unsaturated fatty acids than saturated ones [4]. However, the SB skins had comparatively higher concentrations of saturated fatty acids for both harvests. The PN skins from harvest 1 contained more unsaturated fatty acids while saturated fatty acids increased significantly in the harvest 2 grape skins. Therefore, there might be an increased rate of saturation in fatty acids in the PN grape skin during the ripening process. Interestingly, the SB pulp from the harvest 1 grapes had higher unsaturated fatty acids, while the harvest 2 grape pulps had more saturated fatty acids. However, the PN grape pulp showed an opposite trend. Therefore, the evolution of fatty acid saturation during grape ripening might vary between varieties.

In addition to determining the total lipid content, we also investigated the specific fatty acids that are predominantly present in different grape tissues of both SB and PN. Figure 2 presents the abundance of major fatty acids found in the pulp, seeds and skin of both grape varieties. Stearic acid was the most abundant fatty acid both in the SB and PN grape pulp and skin (Figure 2), while the seeds contained a high amount of linoleic acid (at least >0.1 g/g). This observation was in accordance with previously published studies [4,5]. Among other fatty acids, grape seeds and skin also contained a high concentration of oleic and palmitic acids (>0.05 g/g). Concentrations of other fatty acids present in different grape tissues and varieties were <0.01 g/g. However, even in such small concentrations, these fatty acids play important biological roles during grape development [6].

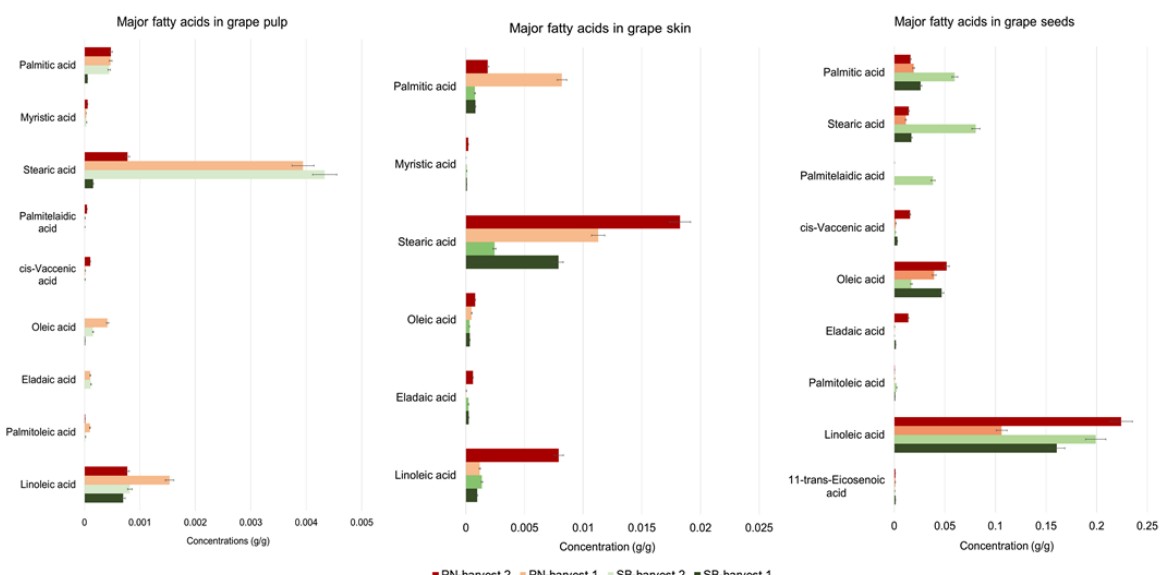

**Figure 2.** Concentrations of major fatty acids in Pinot noir (PN) and Sauvignon blanc (SB) grape pulp, seed and skin. Fatty acid profiling was completed using 10 mg of each sample type.

### 3.4. Total Lipid and Fatty Acid Contents in Sauvignon Blanc and Pinot Noir Grape Juices and Extracts

3.4.1. Effect of Harvest Time on Total Lipids and Fatty Acids in Grape Juices

Table 4 shows the ranges of the total lipids in the SB and PN grapes just after pressing and without any pomace contact. A comparison between the harvest 1 and 2 data shows a significant increase ($p < 0.01$; >30% increase) in lipids in harvest 2 compared with harvest 1 for the SB juices. However, lipid concentrations were not significantly different in the PN grapes harvested at the two time points ($p > 0.05$). Therefore, lipid development in grapes may vary depending on variety, and we indeed found a varietal difference in total lipid contents ($p > 0.05$) as the PN juices from both harvests contained more lipid than the SB juices. A more comprehensive multi-seasonal study on lipids is required to confirm these observations as a seasonal variation is largely observed in grape metabolites [8].

**Table 4.** Total lipids present in Sauvignon blanc and Pinot noir grape juices and the influence of pomace contact on extraction of lipids in the must.

| Sample | Pomace Contact Time (h) | Total Lipids-Harvest 1 (g/L) | Total Lipids-Harvest 2 (g/L) | Change in Lipid Level (%) |
|---|---|---|---|---|
| Sauvignon blanc | 0 | 0.14 (0.01) | 0.22 (0.03) | 36.13 |
| | 24 | 0.41 (0.03) | 0.61 (0.05) | 32.55 |
| | 48 | 0.72 (0.04) | 1.12 (0.08) | 35.84 |
| Pinot noir | 0 | 0.27 (0.03) | 0.33 (0.01) | 16.24 |
| | 72 | 0.23 (0.02) | 0.59 (0.04) | 60.95 |
| | 144 | 0.39 (0.05) | 0.93 (0.06) | 58.02 |

Standard deviations are shown within brackets.

Absolute quantification of 45 different fatty acids ranging from C8 to C24 allowed us to determine the differences observed in the SB and PN grapes harvested at different time points, thus representing variation caused by the different degree of ripeness of the grapes. Not all the fatty acids present in the grape juices were responsible for the variation. In the SB grape juices, at least 30 fatty acids were significantly different ($p < 0.05$) between harvest 1 and 2, while 23 fatty acids were found to be considerably different ($p < 0.05$) in the PN juices. Therefore, we used the 23 most significant ($p < 0.05$) and common fatty acids found in SB and PN to perform a PCA (Figure 3) where principal component (PC) 1 and PC 2 explained more than 57% of the variation. We observed a clear difference in fatty acid composition between the SB and PN juices. In particular, long-chain polyunsaturated fatty acids such as linoleic acid and gamma-linolenic acid were more abundant in the PN juices than SB.

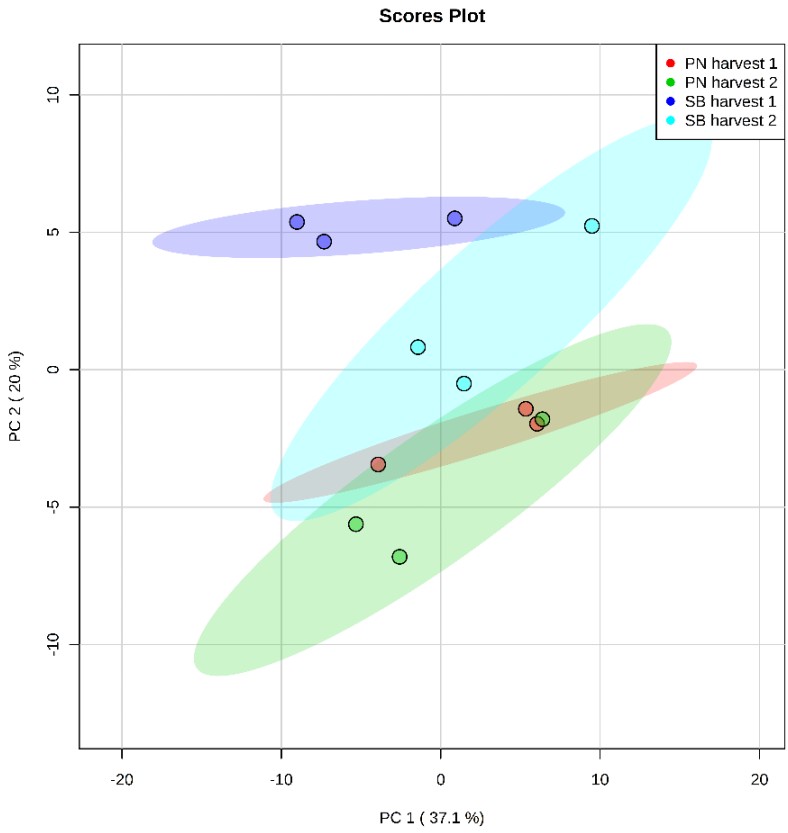

**Figure 3.** Two-dimensional representation of principal component analysis (PCA) using the 23 most significant fatty acids ($p < 0.05$) present in Sauvignon blanc (SB) and Pinot noir (PN) grape juices harvested at two different maturity levels.

While comparing the fatty acid profiles of the two different harvests, we saw a visible difference between SB harvest 1 and 2 that was not prominent for the PN juices (Figure 3). In SB, levels of some saturated fatty acids including palmitic, stearic and pentadecanoic acids increased significantly in the harvest 2 juices. The PN juice profiles showed no such trend except that the concentration of lignoceric acid was higher in harvest 2 than harvest 1 juices. These data indicate that there is a varietal difference in fatty acid developments in grape juices. Moreover, variation in fatty acids between harvests/maturity also may occur for some varieties, but not for all.

3.4.2. Effect of Pomace Contact Time on Lipid Extraction and Fatty Acids in Grape Must

Our next objective was to explore how pomace contact influences lipid extraction in grape juices. Although pomace contact is an uncommon practice for commercial SB wine production, a short contact time (24–48 h) was applied. As shown in Table 3, lipid concentrations increase 2.5- to 5-fold at 24 and 48 h when compared with juices without any pomace contract for both harvests. In addition, a substantial increase in total lipid in the SB must was observed when comparing harvest 1 and 2 after the pomace contact. As the grape juice matrix can be significantly different at different maturity levels, harvest 2 most probably provided more suitable lipid extraction conditions.

To make this study relatable to commercial practice, we used an extended pomace contact for the PN juices. The PN must data showed a different trend, particularly for harvest 1 (Table 3). There was a slight decrease in the total lipids after 72 h pomace contact in the harvest 1 samples, and the reasons for this are unknown. However, lipid concentration increased 1.5-fold after 144 h when compared with the juices without pomace contact (Table 3). Similar to the SB must, comparison between the harvest 1 and 2 data show an increased rate of extraction of lipids after both 72 and 144 h pomace contact, which reinforces our theory on the role of the juice matrix.

Lipid extraction in the SB must was comparatively higher than in PN even though the PN pomace contact time was much longer. We can again relate this observation to the differences in the juice matrices. The SB juice is generally more acidic, which might have influenced the lipid extraction. Additionally, the SB seeds are usually larger than the PN seeds. Moreover, the PN skins contain a larger number of polyphenols and anthocyanins than SB, and thus extracted more of these secondary metabolites than lipids. Therefore, we speculate that the SB cold soak conditions may allow a better lipid extraction than PN. These data confirm that lipids in commercially produced grape musts (especially for white varieties) can be increased by extending the duration of pomace contact depending on the style of wines to be produced.

As shown in Table 3, lipid contents significantly increase in the must due to the prolonged skin/pomace contact. We observed a similar trend in fatty acids. Figure 4 presents the heatmaps that show those fatty acids that changed significantly ($p < 0.05$) due to the pomace contact. Particularly for SB, levels of more fatty acids (15) increased with time during pomace contact than PN. Only two fatty acids, 7-trans-nonadecanoic and myristoleic acids, were more abundant in juices with no pomace contact. There is a possibility that these fatty acids may have undergone some degradation or transformation process during pomace contact, thus contributing to the development of other fatty acids [7]. For instance, the concentration of myristelaidic acid increased over the time of pomace contact while myristoleic acid decreased. These two fatty acids are closely related to each other (C14:1) and might have just transformed during pomace contact. Some of fatty acids also could be consumed and/or produced by natural microorganisms present in the pomaces and musts [15,42].

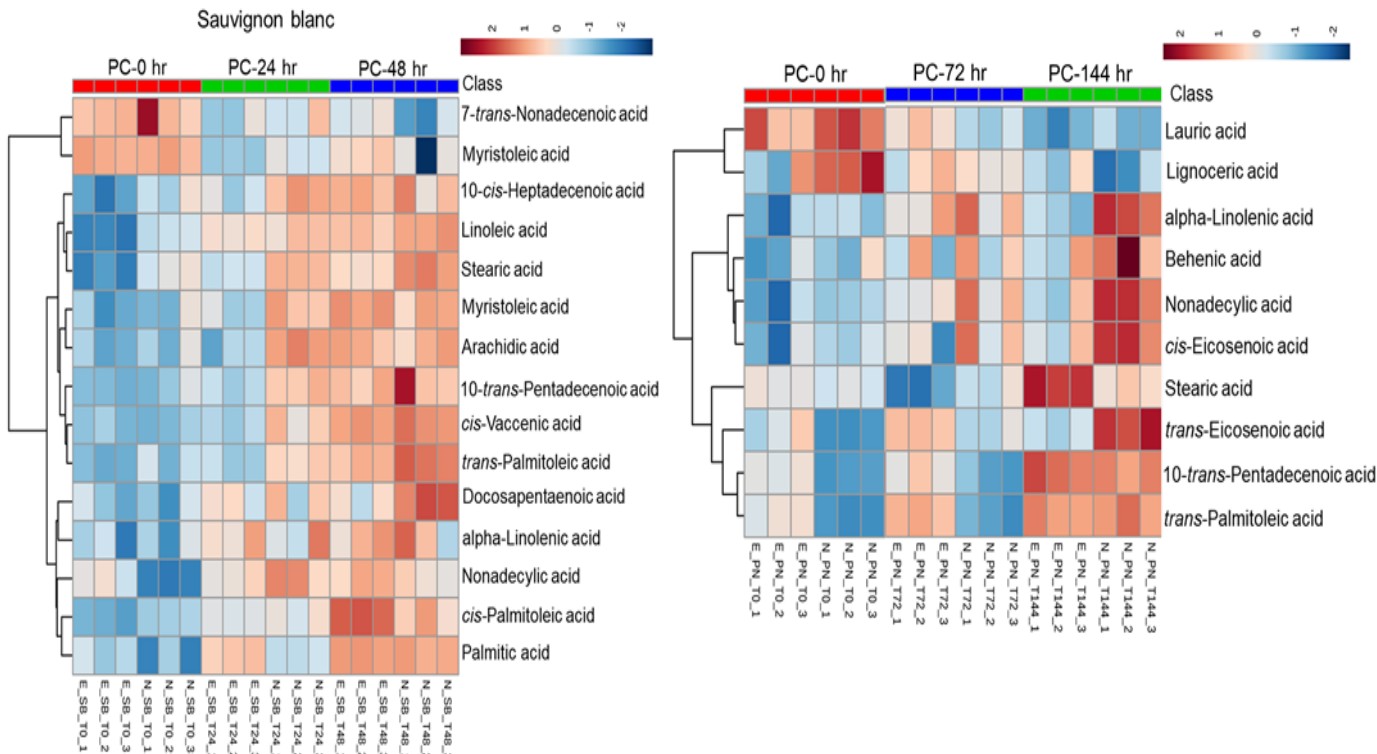

**Figure 4.** Heat map showing the abundance of most significant fatty acids (*p* < 0.05) that were affected by pomace contact (PC) in Sauvignon blanc (SB) and Pinot noir (PN) must. Here: E = Harvest 1, N = Harvest 2, T = time of pomace contact (PC).

*3.5. Effect of Ethanol Concentration Relevant to Fermentation on Lipid Extraction from Grape Pomace*

This part of the experiment was designed to simulate different stages of fermentation to observe how different ethanol concentrations affect lipid and fatty acids extraction. We replaced the must juices with two different aqueous ethanol solutions (9 and 13% *v/v*) to contrast lipid extraction in the absence of ethanol (0%, water only) at 72 and 144 h. Different temperatures were applied to the extractions to reflect commercial winemaking conditions, with the SB extracts incubated at 15 °C and PN at 24 °C (Figure 1). Table 5 shows the summarized data and indicates that lipid extraction increased linearly with the ethanol concentration and also with time. Similar to our observation shown in Table 3, extraction of lipids was greater from the SB pomaces (as high as 3.11 g/L at 144 h) than PN (1.85 g/L at 144 h). Although lipid extraction was expected be better at the higher temperature used for PN, our data proved it otherwise. However, more work is needed to determine the exact reasons behind this observation. As indicated earlier, spontaneous fermentation occurred in some of the extraction conditions, particularly when we used 0 and 9% ethanol, suggesting that some of the lipids may have been originated from the native microbes. This is also evident in commercial winemaking, where winery microbes can contribute to extraction of different metabolites (including flavor precursors) during the maceration process [43,44].

**Table 5.** Effect of ethanol concentrations on lipid extraction from Sauvignon blanc and Pinot noir grape pomace. Standard deviations are shown within brackets.

| Variety | Sauvignon Blanc | | | | | | | | |
|---|---|---|---|---|---|---|---|---|---|
| Ethanol (% *v/v*) | 0 | 0 | Increase (%) | 9 | 9 | Increase (%) | 13 | 13 | Increase (%) |
| Pomace contact time (h) | 72 | 168 | | 72 | 168 | | 72 | 168 | |
| Total lipids-harvest 1 (g/L) | 0.70 (0.04) | 1.09 (0.05) | 35.51 | 0.86 (0.16) | 1.80 (0.19) | 52.02 | 1.04 (0.18) | 1.75 (0.14) | 40.43 |
| Total lipids-harvest 2 (g/L) | 0.69 (0.04) | 1.14 (0.10) | 39.45 | 0.99 (0.10) | 1.84 (0.12) | 45.96 | 1.53 (0.09) | 3.11 (0.30) | 50.93 |
| Variety | Pinot Noir | | | | | | | | |
| Ethanol (% *v/v*) | 0 | 0 | Increase (%) | 9 | 9 | Increase (%) | 13 | 13 | Increase (%) |
| Pomace contact time (h) | 72 | 168 | | 72 | 168 | | 72 | 168 | |
| Total lipids-harvest 1 (g/L) | 0.38 (0.02) | 0.85 (0.06) | 54.62 | 0.48 (0.08) | 1.06 (0.13) | 54.15 | 1.05 (0.08) | 1.60 (0.17) | 34.65 |
| Total lipids-harvest 2 (g/L) | 0.54 (0.03) | 0.89 (0.07) | 38.60 | 0.89 (0.05) | 1.24 (0.04) | 28.51 | 1.44 (0.02) | 1.85 (0.14) | 22.09 |

We also monitored the changes in the fatty acid profiles during ethanolic extractions of the SB and PN pomaces. For both the SB and PN pomaces, most of the fatty acids were better extracted when ethanol was present, which was expected. However, some of the fatty acids were extracted better when there was no ethanol including linoleic acid for both SB and PN and octanoic, trans-vaccenic acids only for SB. Therefore, we assume that the fatty acid composition of grape must can not only be changed through prolonged pomace contact, but also can be manipulated during the winemaking condition, particularly when ethanol starts to be produced during the fermentation. These results also indicate that different types of fatty acids are available at the different stages of fermentation with the progress of ethanol production. Moreover, fatty acids and lipid components from yeast cells would also contribute to the lipid availability from mid- and late-fermentation, particularly when exogenous lipid sources are all consumed [15]. Therefore, this knowledge would assist us in developing strategies to manipulate fatty acids and lipids during winemaking, thus influencing the final aroma bouquet of wines as many of these fatty acids serve as a pre-cursor for the formation of ethyl and acetate esters [10].

*3.6. Correlation of Lipids and Fatty Acids with Major Oenological Parameters*

We performed a correlation analysis to investigate if lipids and major fatty acids found in the SB and PN juices have any relationship with major amino acids and other oenological parameters. Figure 5 presents the top 25 features that positively or negatively correlated with the total lipids in the juices. While primary amino acids, YAN and ammonium show strong positive correlation with the total lipids, individual amino acids exhibit negative correlation. The reasons for this remain unknown at this point and warrant further investigation.

Among fatty acids, stearic and linoleic acids also show positive correlation with these ripeness parameters (Figure 6). These two fatty acids are also major fatty acids present in different NZ grape varieties [8]. Although stearic acid strongly correlated with the total lipids (Figures 5 and 6), linoleic acid did not show such trend. We assume that as a polyunsaturated fatty acid, linoleic acid is more prone to oxidation and other chemical changes during the lipid extraction time, while stearic acid is more stable. Therefore, stearic acid has more potential to be a ripeness indicator and a simple user-friendly test to determine stearic acid can be developed to be applied for the determination of berry ripeness.

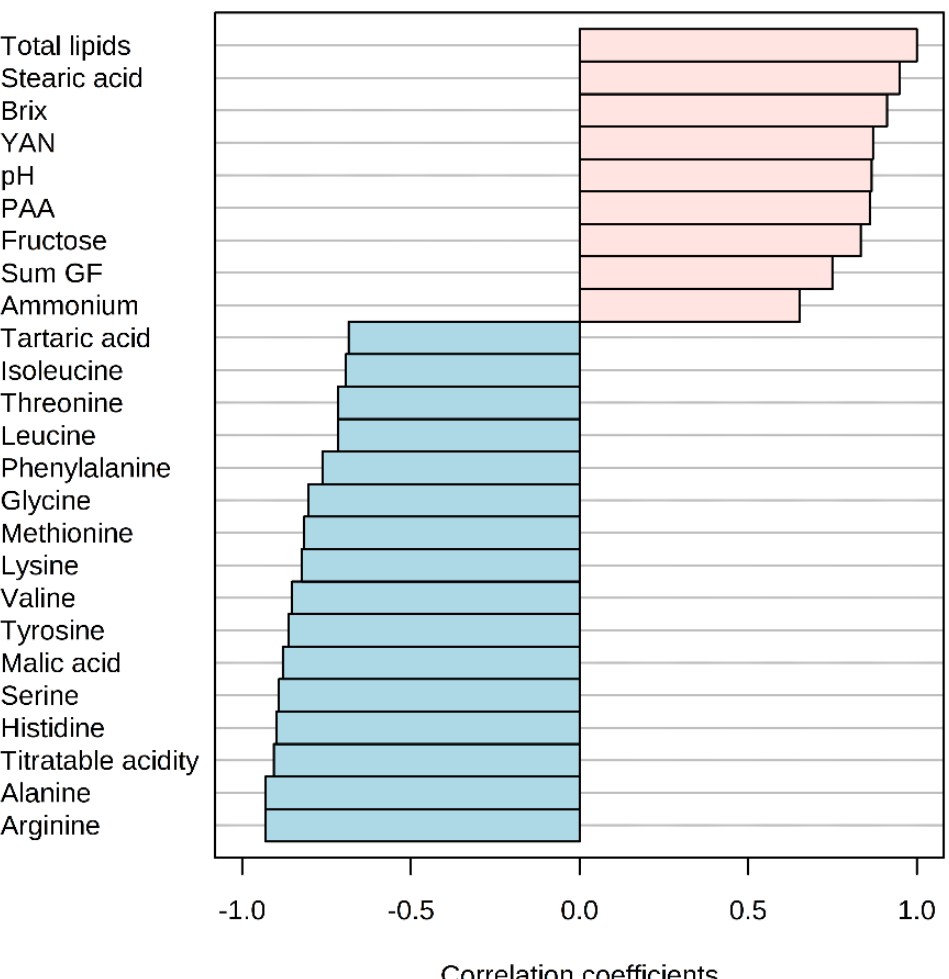

**Figure 5.** A pattern hunter plot showing the correlation of total lipids with different oenological parameters and nitrogenous compounds analyzed in the Sauvignon blanc and Pinot noir grape juices. Pink bars represent the positive correlation (Pearson r > +0.50) while blue bars show the negative correlation (Pearson r > −0.50). YAN = yeast available nitrogen, PAA = primary amino acids, GF = glucose + fructose.

The relationships between lipids/fatty acids and different ripeness parameters have largely been overlooked and while searching the literature we found them only in a few studies. For instance, Bauman et al. [45] investigated the lipid composition and fatty acid distribution of Concord grapes over four different stages of maturation. They reported that crude lipid content was highest at *véraison* (0.23%) while neutral lipids increased and polar lipids decreased during maturation, indicating that lipid composition of grapes evolves at different stages of grape development. Another study published by Le Fur et al. [46] investigated the changes in phytosterols (ß-sitosterol, campesterol, stigmasterol and lanosterol) in grape skins during the last stages of ripening of Chardonnay grape variety in Burgundy. Barron and Santa-María [47] investigated the relationship between triglycerides and different ripeness and energy indices. Their results indicated a strong correlation of different triglycerides species with energy indices while showing a moderate correlation with different ripeness indices. In a more recently published study using comprehensive lipidomics approach, Masuero et al. [11] reported positive and negative correlation with certain lipid classes with total soluble solids, indicating a relationship among grape ripeness with lipids. However, they did not determine the correlation between

the total lipids or major fatty acids with grape ripeness. Our data, therefore, provide a novel insight and we hypothesize that lipid concentration increases with grape maturation, which is related to the different stages of ripening. Thus, lipids could be another parameter for determining grape ripeness alongside sugars/acids. Using this information, further research could be undertaken to develop either a colorometric industry friendly test or non-destructive near infrared spectroscopy (NIRS) method to determine the total lipids or a specific lipid biomarker linked with ripeness. This would provide the industry an important tool to determine grape quality prior to harvest. This in turn would provide vital knowledge for winemakers to select appropriate yeast strains to produce wines with good aromatic quality.

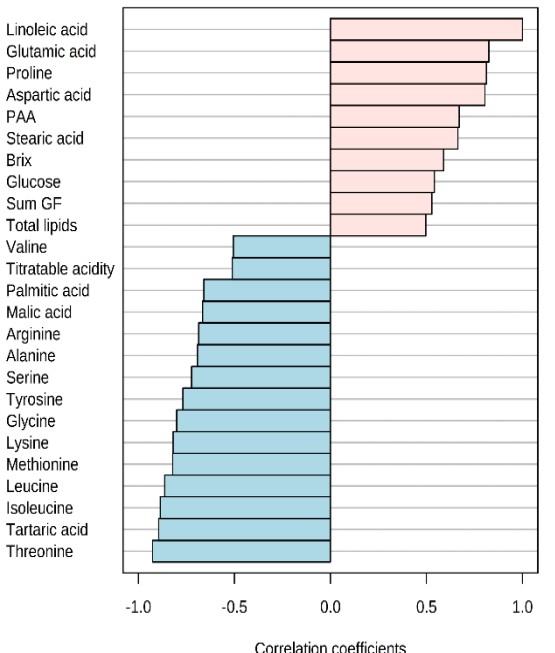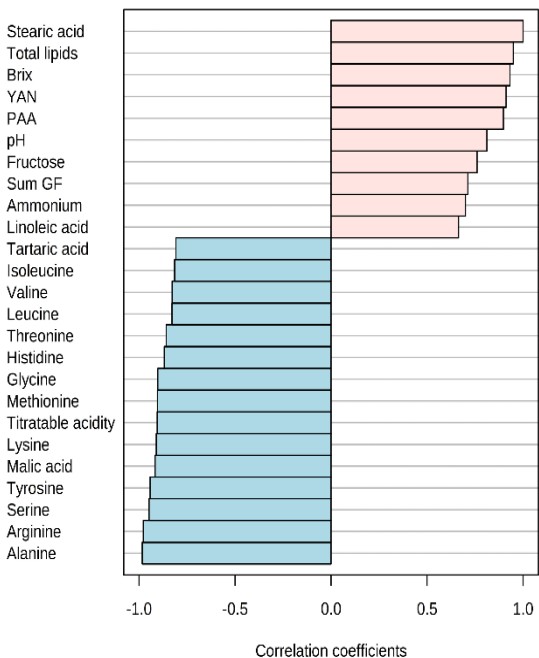

**Figure 6.** Pattern hunter plots showing the correlation of linoleic and stearic acids with different oenological parameters and nitrogenous compounds analyzed in the Sauvignon blanc and Pinot noir grape juices. Pink bars represent the positive correlation (Pearson r > +0.50) while blue bars show the negative correlation (Pearson r > −0.50). YAN = yeast available nitrogen, PAA = primary amino acids, GF = glucose + fructose.

As is evident from the available published data, lipids and fatty acids are less studied classes of compounds in oenology. Most of the research in this space has focused on the role of fatty acids in aroma development and yeast metabolism [12,13,15,19,21,48–51]. Research by Sherman [2] already demonstrated the prospective role of unsaturated fatty acids as predictors of perceived PN wine body as lipids persist in wines after fermentation, thus contributing to wine sensory properties. Therefore, more research is needed to extend different aspects of utilization of lipid molecules in winemaking and also in grape growing. There is a large knowledge gap and we found no multi-season study that focused on the lipid developments during grape growing and how lipids/fatty acids evolve during different stages of winemaking.

## 4. Conclusions

In this study, we generated some novel insights on the evolution of lipid and fatty acids in grapes at different stages of maturity by analyzing grape juices, grape tissues and ethanolic extracts. We found that the total lipids and fatty acid composition vary between harvests at different stages of ripeness depending on the grape variety. We

observed a strong correlation between lipids/major fatty acids and other commonly used ripeness parameters including total soluble solids, sugars, titratable acidity and YAN, thus indicating a potential role of lipids as another ripeness parameter. Our data also showed that lipid concentrations increased significantly because of prolonged skin/pomace contact depending on the grape variety. Moreover, lipid and fatty acid extraction increased linearly with the ethanol concentration and time of pomace contact while extraction of lipids and fatty acids was greater in SB maceration conditions than PN, suggesting a matrix effect. As our data are based on only one season, observations might not be conclusive and there is a need for a comprehensive multi-season study to confirm these data-generated hypotheses. Research should also be carried out to determine the effect of different vineyard management practices on the developments of lipids in grapes. If this type of research is successfully undertaken, the wine industry will ultimately benefit from the knowledge generated to produce diverse styles of wines via manipulating different lipid classes and fatty acids during grape growing and winemaking.

**Author Contributions:** Conceptualization, E.S. and F.R.P.; methodology, E.S., M.Y., F.G., E.Z., S.G. and F.R.P.; formal analysis, E.S., K.W.B. and F.R.P.; data curation, E.S. and F.R.P.; writing—original draft preparation, E.S. and F.R.P.; writing—review and editing, E.S., M.Y. and F.R.P.; project administration, F.R.P.; funding acquisition, F.R.P. All authors have read and agreed to the published version of the manuscript.

**Funding:** The authors declare that this study received funding from New Zealand Lighter Wines Programme (P/471777/04) co-funded by New Zealand Winegrowers, and the Ministry of Primary Industries' Primary Growth Partnership (PGP) venture. The funder was not involved in the study design, collection, analysis, interpretation of data, the writing of this article or the decision to submit it for publication.

**Institutional Review Board Statement:** Not applicable.

**Informed Consent Statement:** Not applicable.

**Data Availability Statement:** Access to data and database can be requested by contacting the corresponding author.

**Acknowledgments:** We are thankful to Gerald Hope of the Marlborough Research Centre for kindly providing Sauvignon blanc grapes from their Rowley Vineyard and to Nigel Sowman from Dog Point for generously providing Pinot noir grapes. Acknowledgment is due to Victoria Raw, Rob Agnew, Claire Grose, Lily Stuart, Tanya Rutan and Rachel Bishell for their help during the study. We also thank Grant Morris, Megan Jones, Kevin Sutton and Plant and Food's Science Publication Unit for their help with manuscript.

**Conflicts of Interest:** The authors declare no conflict of interest.

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
