# Peer review of "Total Lipids and Fatty Acids in Major New Zealand Grape Varieties during Ripening, Prolonged Pomace Contacts and Ethanolic Extractions Mimicking Fermentation"

_fermentation, doi:10.3390/fermentation9040357_

Round 1

Reviewer 1 Report

The article is well return such that there's little to no revision required. The article was enjoyable to read. There was a nice flow on the manuscript. I am very much impressed with the author's analytical work.

All what's left for me is to say well done to the authors!

Author Response

We thank the reviewer for such positive comments on our manuscript and we are really glad to hear that the reviewer enjoyed reading the manuscript. 

Reviewer 2 Report

In this research work, fatty acid profiles and total lipid content in two of NZ’s major grape varieties were determined. However, I can send some observations to improve the manuscript:

Introduction:

Pag. 1, line 38-41. I suggest you update references 5 and 6. I think you can find more recent information.

Pag. 2, line 65-67. I suggest you update reference 10. I think you can find more recent information.

Materials and Methods:

Pag. 4, line 162-163. I suggest you update reference 27. I think you can find more recent information.

Pag. 5, line 179-182. I suggest you update reference 28. I think you can find more recent information.

Pag. 5, line 207-208. I suggest you update reference 32. I think you can find more recent information.

Results and discussions:

Table 1 shows the results of reducing sugars, but I don't see their methodology. Can you clarify this point?

Pag. 16, line 551-550. I suggest you update references 50 to 53. I think you can find more recent information.

I congratulate the authors, it is a good document, the quality is very high, it is very well discussed.

Author Response

Thank you so much for your feedback. Please see attached our responses. 

Reviewer 3 Report

The authors deepened the knowledge about the composition of lipids and fatty acids in the two main grape varieties of New Zealand.

This information may be relevant for producers, giving them more tools to obtain hight quality wines.

My comments are in the PDF file, marked in yellow along with the text.

Author Response

Thank you for your constructive feedback on our manuscript. Please see attched our responses. 
